# The FineMotion entry to the GENEA Challenge 2023: DeepPhase for conversational gestures generation

Vladislav Korzun
korzun@phystech.edu
Moscow Institute of Physics and
Technology
Moscow, Russia
Tinkoff
Moscow, Russia

Anna Beloborodova
beloborodova.as@phystech.edu
Moscow Institute of Physics and
Technology
Moscow, Russia
Tinkoff
Moscow, Russia

Arkady Ilin
arkady.ilin@skoltech.ru
Skolkovo Institute of Science and
Technology
Moscow, Russia
Tinkoff
Moscow, Russia

## ABSTRACT

This paper describes FineMotion's entry to the GENEA Challenge 2023. We explore the potential of DeepPhase embeddings by adapting neural motion controllers to conversational gesture generation. This is achieved by introducing a recurrent encoder for control features. We additionally use VQ-VAE codebook encoding of gestures to support dyadic setup. The resulting system generates stable realistic motion controllable by audio, text and interlocutor's motion.

## CCS CONCEPTS

• **Computer systems organization** → **Embedded systems**; *Redundancy*; Robotics; • **Networks** → Network reliability.

## KEYWORDS

embodied agents, neural networks, gesture generation, social robotics, deep learning, phase manifold

**ACM Reference Format:**
Vladislav Korzun, Anna Beloborodova, and Arkady Ilin. 2023. The FineMotion entry to the GENEA Challenge 2023: DeepPhase for conversational gestures generation. In *INTERNATIONAL CONFERENCE ON MULTIMODAL INTERACTION (ICMI '23), October 9–13, 2023, Paris, France.* ACM, New York, NY, USA, 6 pages. https://doi.org/10.1145/3577190.3616119

## 1 INTRODUCTION

The automatic generation of conversational gestures for 3D human models is one of the most opportune problems in character animation. It can be used to simplify video game production and increase the realism of characters' movements. Furthermore, as visual assistants or VTubers are becoming more popular, the demand for realistic gestures for embodied virtual agents is also growing.

The task of automatic gesture generation from speech has got several promising solutions. During GENEA Challenge 2022 [25] one of the approaches was rated even better than real motion capture data by motion quality [27]. However, the task at hand is becoming more complicated year by year.

The current GENEA Challenge 2023 [15] considers a dialogue setup. Thus, the participants' systems should not only consider input speech but also the conversation partner's behaviour. As well as in the previous year «Talking With Hands 16.2M» dataset [16] was used, but now each sample contains two sets of motion, audio and text for the main agent and the interlocutor.

In relative tasks of condition-based motion generation [23] and character controllers [26] researchers propose slightly different approaches, that could also benefit conversational gestures generation. One of the most promising approaches for animation representation was presented in [19]. Taking into account that motion curves could be considered as periodic functions, they could be decomposed via Fourier Transform to obtain high-level features.

Thus, we decided to examine the phase manifold formed by DeepPhase's Periodic AutoEncoder in conversational gesture generation. In order to properly address the dyadic setup of the challenge, we implemented additional interlocutor gesture representation based on VQ-VAE codebook encoding. Evaluation [15] showed that our system generates realistic motion which is statistically suitable for the interlocutor's behaviour. However, our system showed poor results on appropriateness for speech, which suggests the need for further development. Our code along with video examples of generated motion is publicly available[1] to help other researchers reproduce our results.

Our paper is organized as follows: Section 2 gives an appropriate overview of related work; Section 3 describes our approach generally; Section 4 details generator model input and output format; Section 5 gives results from the evaluation and discusses our results; and Section 6 is for the conclusion.

## 2 RELATED WORK

In this section, we give a general overview of recent conversational gesture generation approaches. Then we describe some existing approaches for solving close tasks, that inspired our solution.

### 2.1 Conversational gestures generation

The task of conversational gestures generation has been advancing for several years. Starting from window-based frame-by-frame generation [13] end-to-end approaches lead to auto-regression [14]. Later, the GENEA Challenge 2022 offered many successful systems. Some of them are based on recurrent models [4, 6, 24], and some

---

[1]https://github.com/FineMotion/GENEA_2023

even utilise GPT-like large architectures [18], but the most successful hybrid approach was presented in [27], where authors use the graph-based model to transfer between short clips.

Slightly weaker results were shown by clear auto-regressive approaches [11, 12], that faced the main shortcoming of such architectures - converging to mean pose. In [12] as well as in [14] authors tried to overcome this problem by adding different teacher-forcing techniques to force models first to extract appropriate audio representation. However, auto-regressive approaches have shown significant success without such techniques in a different task: character controllers.

## 2.2 Character controllers

The task of creating automatic character controllers is related to locomotion movements [8]. The controlled character should move joints with respect to the environment and user input. Many data-driven character controller approaches use a mixture-of-experts [10] framework, for example, Mode Adaptive Neural Networks (MANN) [26].

Later, the MANN model was improved with local phases [20]. Local phases are computed as a derivative from block function containing binary states of whether bone contacts the object/environment. The efficiency of the proposed approach was demonstrated in creating a neural motion controller for a basketball game, where the block function represented a player's contact with the ball or the floor.

Finally, in [19] the unsupervised approach for automatic phase extraction was suggested. The proposed Periodic AutoEncoder extracts periodic features from motion curves after training on unstructured motion datasets. The architecture utilizes a temporal convolutional autoencoder [9] additionally applying real Fast Fourier Transform to each channel of latent space. The obtained periodic features then were used to train the motion controller as before showing the capability of extracted features.

## 2.3 Text-to-Gesture Animation Generation

The task of generating human gesture animations from textual prompts involves generating expressive and natural-looking gestures that correspond to a given textual input. For example, in the work of [7] the authors suggest jointly encoding gestures, text and images into a single latent space using Contrastive Language-Image Pretraining (CLIP) [2]. Also, in GestureDiffuCLIP [21] the authors combined the power of CLIP and diffusion models to generate realistic and diverse gesture animations from text. To enable the encoding and decoding of gestures, the Vector Quantized Variational Autoencoder (VQ-VAE) [1] was used. Additionally, VQ-VAE has proven to be a valuable tool beyond text-to-gesture generation. In the context of conversational gestures, recent research [18] and [22] applied the VQ-VAE to encode and decode gestures, achieving improved gesture generation performance.

## 3 SYSTEM OVERVIEW

Our approach follows the original DeepPhase paper [19]. It contains two main stages: training Periodic AutoEncoder to extract phase features and building neural motion controller upon extracted phases. The motion controller is based on a mixture-of-experts framework

also mentioned in the DeepPhase paper with some ideas from previous author's work [20]. The main difference between our system and those mentioned above is that we use an auxiliary recurrent Control Variables Encoder to guide motion by audio, text and interlocutor's motion instead of the user's input. Apart from that, we trained an additional encoder for the interlocutor's motion and supplemented control features with the obtained latent representation.

## 3.1 DeepPhase embeddings

To prepare the phase manifold we follow the proposed pipeline from [19] exactly. To train Periodic AutoEncoder (PAE) we first extract positions from the main agent's motion data. We use all motion files, but extract positions for 26 joints, including world root and excluding fingers. Then we calculate joint position velocities and smooth them via Butterworth Filter [3].

The training configuration of PAE is as follows: training sample contains 61 frames and covers a 2-second window with 26*3 channels. The number of latent channels (phases) is equal to 8, following the dancing pipeline from the official repository[2]. The number of intermediate channels is equal to the number of joints. The model is trained during 150 epochs with batch size equal to 512 and AdamW optimizer with Cyclic Learning Rate Scheduler with Restarts [17] with weight decay and learning rate both equal to 10e-4, restart period equal to 10, multiplier equal to 2 and cosine policy.

The obtained model extracts phase features as in the original paper. From each time window $t$ it extracts amplitude ($A$), frequency ($F$), offset ($B$) and phase shift ($S$). $A, F, B, S \in \mathbb{R}^M$, where $M$ - number of latent channels (or phases). Phase manifold $\mathcal{P} \in \mathbb{R}^{2M}$ for frame $t$ is computed by

$$\mathcal{P}_{2i-1}^{(t)} = A_i^{(t)} \cdot sin(2\pi \cdot S_i^{(t)}), \quad \mathcal{P}_{2i}^{(t)} = A_i^{(t)} \cdot cos(2\pi \cdot S_i^{(t)}). \quad (1)$$

To obtain phase features $\mathcal{P} \in \mathbb{R}^{T \times 2M}$ from motion with length $T$ we just extract the phase manifold from the sliding window, i.e. $\mathcal{P} = \{\mathcal{P}^{(t)} | t \in [1, T]\}$. In order to illustrate the periodicity of extracted phase features the Figure 1 shows them separated by latent channel on a 10-second sample.

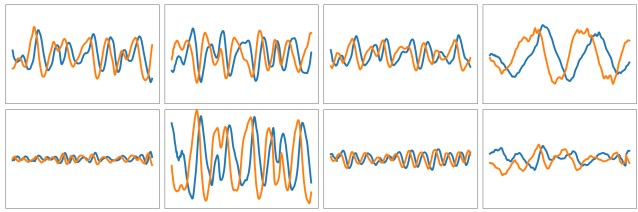

**Figure 1: Extracted phase features example**

Due to the fact that PAE is trained on joint velocities, obtained phases can not be used as intermediate representations of motions instead of original data to train motion generator. The problem lies in the difficulty of converting joint positions into joint rotations without the introduction of kinematic constraints. To overcome this we also tried to train PAE on joints rotations. Unfortunately, obtained phase manifold does not look like periodic function as

---

[2]https://github.com/sebastianstarke/AI4Animation

before. PAE trained on angle velocities could theoretically shows better results, but we decided to stop on phase manifold trained on joint velocities.

## 3.2 Generation model

Our motion generation model extends the mixture-of-experts framework from [19]. It contains two feedforward neural networks: Gating Network and Motion Prediction Network. The model's notation follows [20].

The Gating Network is built upon a stack of linear layers with ELU[5] activations between them. It takes phase features and predicts weights for experts. In our case, there are 8 experts. Then, the Motion Prediction Network uses these weights to make linear combinations over experts. The Motion Prediction Network itself consists of several "Expert Layers" with ELU activations between them. Each of layer $E$ uses experts weights $\alpha = \{\alpha_i, i \in [1, N]\}$ and input $x$ as follows:

$$E(x, \alpha) = \sum_{1}^{N} \alpha_i(W_i x + b_i) \qquad (2)$$

where $W_i \in \mathbb{R}^{h \times m}$ and $b_i \in \mathbb{R}^h$ are weights and biases respectively with $m$ and $h$ being input and output dimensions respectively. As in the original DeepPhase repository, the number of "Expert layers" as well as the number of linear layers on the Gating Network is equal to 3.

## 3.3 Control Variables Encoder

Initially, the input and output data formats were similar to [20]. However, significant changes were introduced. As control variables input, we use a similar time window of audio features. But the more control features like text and interlocutor's pose we added, the larger the control variables vector would become. So we decided to add an additional recurrent encoder of control features based on Bi-directional GRU over the FeedForward Highway as in [12] to shorten this vector. It takes time-window features around the current frame and returns the output vector from RNN corresponding to the considered frame.

## 3.4 Interlocutor Gesture Encoder

*Model.* To effectively respond to the gestures of the interlocutor, our model leverages the Interlocutor Gesture Encoder, a crucial component based on the VQ-VAE framework from [1]. This model showed good results in gesture coding, as shown in [18] and [22]. The Interlocutor Gesture Encoder enables us to encode high-quality representations of gestures into compact vectors.

For better learning, we have added improvements such as exponential codebook smoothing and discarding unused vectors, as suggested in the original article.

*Data processing.* To train the VQ-VAE model, we segment gestures into gaps according to the bits in the audio. This idea was proposed in the [22]. The authors proposed dividing gestures into segments that align with the rhythmic structure of the audio, as it is believed to capture the salient aspects of the gestures. The maximum number of frames in one gesture's sample with this approach is equal to 18. This approach has shown promising results

in capturing the temporal dynamics and synchronizing gestures with the corresponding audio cues. Building upon this concept, we adopt a similar data processing strategy in our study to leverage the benefits of aligning gestures with the rhythmic elements of the audio. During training, the network is fed with only those gesture samples from both partners in which at least one conversational partner was speaking. Each selected sample corresponds to the speaker's audio bits. During inference, we feed only interlocutor's gestures corresponding to the active speaking person's audio bits. In order to determine the moments of speech, we use a text transcript. If there is no active speaker at the moment, main agent's audio bits are chosen for guidance.

*Training.* We train the VQ-VAE model with codebook size 2048. The dimensional of codebook vectors was 256. Codebook occupancy reaches 70%. The model was trained over 152 epochs.

*Inference.* To feed the interlocutor's gestures into the main model, we split the interlocutor's audio into bits, then we extract vectors for each sample. After that, we duplicate each vector to the size of a bit. Thus, we get the number of vectors equal to the number of frames in the original gesture.

## 4 GENERATOR INPUTS AND OUTPUTS

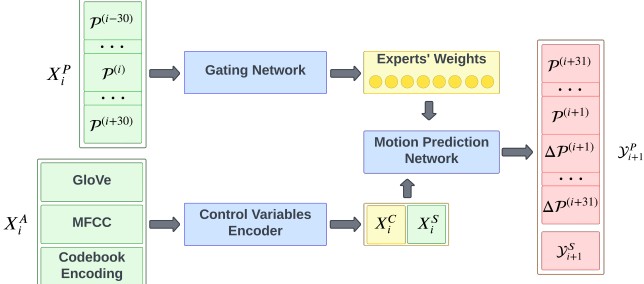

**Figure 2: Generator model**

The overall system is illustrated in Figure 2. The model takes the information from the current frame and predicts the next frame. We use a notation of a time series window similar to [20], i.e. $\mathcal{T}_{t_0}^{t_1}$ represents features collected within a time window $t_0 \leq t \leq t_1$. Following is the description of the final data formats.

*Inputs.* Generator's input consists of 3 components $X_i^S, X_i^A, X_i^P$.

**Character state** $X_i^S$ on $i$-th frame consists of concatenated joints rotations and velocities. We also initially used joint positions, but we observed that the model is more stable without them. We represent joint rotations via 6D continuous representation from [28] to eliminate cases when Euler's angles have values equal to 0 or 180 degrees. Joint velocities were preliminary smoothed as in the PAE training routine. It's also worth mentioning that character state and phases were preliminary normalized.

**Control variables** $X_i^A$ are time-window $\mathcal{T}_{-1s}^{1s}$ features around the current frame, which is passed to Control Variables Encoder to obtain one control vector $X_i^C$, which will be concatenated with character state as the main input to Motion Prediction Network. As initial control features, we extract 26 MFCCs from audio, GloVe embedding of size 50 and obtained codebook encoding from VQ-VAE with respect to motion frame rate which is equal to 30 FPS.

To align text and interlocutor's features we distribute them evenly within frames corresponding to time span. We also tried other combinations, including interlocutor's speech, but they showed less stable results. We decided to make the dimension of $X_i^C$ equal to $X_i^S$.

**Motion Phases** $X_i^P = \Theta_i \in \mathbb{R}^{2K\mathcal{T}}$ are extracted phase features via PAE uniformly sampled from time-window $\mathcal{T}_{-1s}^{1s}$ and concatenated into one vector, i.e. $\Theta_i = \{\mathcal{P}^{(i-30)}, \ldots, \mathcal{P}^{(i-5)}, \mathcal{P}^{(i)}, \mathcal{P}^{(i+5)}, \ldots, \mathcal{P}^{(i+30)}\}$ considering that 13 frames are sampled in the window.

*Outputs.* Our Motion Prediction Network output contains only 2 components: the next frame **character state** $Y_{i+1}^S$, which is similar to input one, and future **motion phases** $Y_{i+1}^P = \{\Theta_{i+1}, \Delta\Theta_{i+1}\}$ containing not only phases, but phases' velocity for time-window $\mathcal{T}_{0s}^{1s}$ with respect to frame $i+1$, i.e $= \Theta_{i+1} = \{\mathcal{P}^{(i+1)}, \mathcal{P}^{(i+6)}, \ldots, \mathcal{P}^{(i+31)}\}$ with 7 frames total.

*Training.* The model is trained to predict the next frame based on the current frame, it does not use outputs from the previous step - every frame is taken from the dataset directly and is processed independently. All parts of the generator are trained simultaneously end-to-end during 50 epochs with batch size equal to 2048 and a default Adam optimizer with a learning rate equal to 10e-4. The hidden sizes of the Gating Network and the Motion Prediction Network are 64 and 1024 respectively.

*Inference.* Finally, during inference, our model predicts the next frame based on the previous one and follows an auto-regressive fashion. We also blend phases between iterations, before passing them to the next step: $\Theta'_{i+1} = \lambda\Theta_{i+1} + (1 - \lambda)(\Theta_i + \Delta\Theta_{i+1})$ with $\lambda = 0.5$.

## 5 RESULTS AND DISCUSSION

As in previous challenges, organizers provided a comprehensive human evaluation of participating systems[15]. This time 3 main subjective measures are considered: human likeness, appropriateness to speech and appropriateness to the interlocutor's behaviour.

Human-likeness estimates the overall quality of generated motion without taking into account the agent's speech or interlocutor's behaviour. Our approach, indexed **SL**, shows competitive results (median score is $51 \in [50, 51]$ in Table 1) indicating the ability of DeepPhase embeddings to maintain periodicity and as a result the realism of predicted motion. Although our model is rated rather well, it does not reach the quality of natural motions or state-of-the-art approaches.

In order to estimate the appropriateness of agent speech, evaluation participants were given two motion clips generated by one model using separate audio samples and tasked to distinguish which of the two motion clips corresponds to the target listening sample. Good models generate motions that participants could easily determine from one another by audio. The main quantity of interest in the appropriateness evaluation is the mean appropriateness score (MAS). Unfortunately, our model provides poor appropriateness results ($0.05 \pm 0.05$ MAS in Table 1). Organizers mentioned (section 3.6 in [15]) that our solution does not statistically differ from chance performance. This leads us to suspect the weakness of used audio and text features.

**Table 1: Summary statistics of studies**

| Condi-
tion | Human-Likeness
Median Score | Agent Speech
MAS | Interlocutor
MAS |
|---|---|---|---|
| NA | $71 \in [70, 71]$ | $0.81 \pm 0.06$ | $0.63 \pm 0.08$ |
| BM | $43 \in [42, 45]$ | $0.20 \pm 0.05$ | $-0.01 \pm 0.06$ |
| BD | $46 \in [43, 47]$ | $0.14 \pm 0.06$ | $0.07 \pm 0.06$ |
| SA | $30 \in [29, 31]$ | $0.11 \pm 0.06$ | $0.09 \pm 0.06$ |
| SB | $24 \in [23, 27]$ | $0.13 \pm 0.06$ | $0.07 \pm 0.08$ |
| SC | $9 \in [\ 9,\ \ 9]$ | $-0.02 \pm 0.04$ | $-0.03 \pm 0.05$ |
| SD | $45 \in [43, 47]$ | $0.14 \pm 0.06$ | $0.02 \pm 0.07$ |
| SE | $50 \in [49, 51]$ | $0.16 \pm 0.05$ | $0.05 \pm 0.07$ |
| SF | $65 \in [64, 67]$ | $0.20 \pm 0.06$ | $0.04 \pm 0.06$ |
| SG | $69 \in [67, 70]$ | $0.39 \pm 0.07$ | $-0.09 \pm 0.08$ |
| SH | $46 \in [44, 49]$ | $0.09 \pm 0.07$ | $-0.21 \pm 0.07$ |
| SI | $40 \in [39, 43]$ | $0.16 \pm 0.06$ | $0.04 \pm 0.08$ |
| SJ | $51 \in [50, 53]$ | $0.27 \pm 0.06$ | $-0.03 \pm 0.05$ |
| SK | $37 \in [35, 40]$ | $0.18 \pm 0.06$ | $-0.06 \pm 0.09$ |
| SL | $51 \in [50, 51]$ | $0.05 \pm 0.05$ | $0.07 \pm 0.06$ |

The addition to this year's challenge is the introduction of the appropriateness metric for the main agent's reaction to the interlocutor's behaviour. The study itself is similar to the previous one with changing interlocutor's motion. It is also conducted while the main agent is silent. Surprisingly, using the interlocutor's motion features yields better results ($0.07 \pm 0.06$ MAS in Table 1) and significantly better than a chance (section 4.7 in [15]).

Overall, our system shows promising results, more on human-likeness and appropriateness for the interlocutor. However, there are ways to improve this approach by adding more compelling audio features or adding teacher forcing to make attention to speech features. Nevertheless, using DeepPhase embeddings allow us to train the model without suffering converging to a rest pose. Additionally, VQ-VAE codebook encoding allowed the resulting solution to accord the dyadic setup of conversation and generate plausible reactions to interlocutor behaviour.

## 6 CONCLUSION

Sharing approaches between different tasks in the domain of motion generation could significantly improve the overall state of the research community. Our system is based on an approach that proved itself as a neural motion controller and showed promising results during evaluation. We assume that using periodic properties of motion could yield improvements in all problems connected with animation. And DeepPhase embeddings are one of the latest and most successful approaches to extract these properties, so we recommend considering them as well as VQ-VAE codebook encoding during the development of future models.

Despite that our system showed relatively good results in the challenge, there is room for improvement. For example, a better speech encoder or additional data filtering could be used. The mixture-of-experts framework could also be extended to work with sequences. Some teacher-forcing techniques could also be applied.

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

## A  PAIRWISE SIGNIFICANT DIFFERENCE FOR APPROPRIATENESS STUDIES

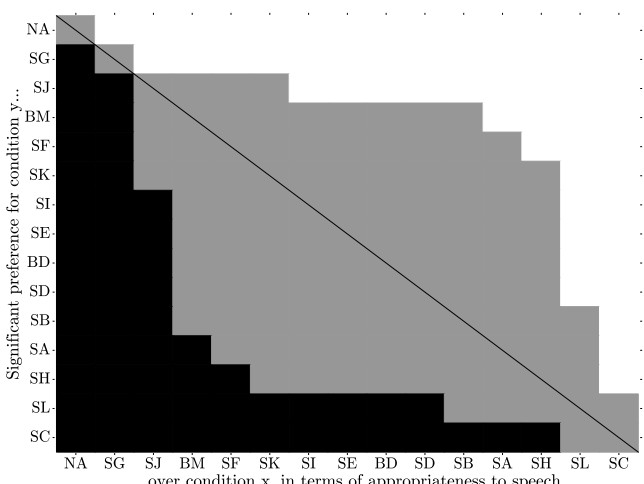

**(a) Appropriateness for agent speech**

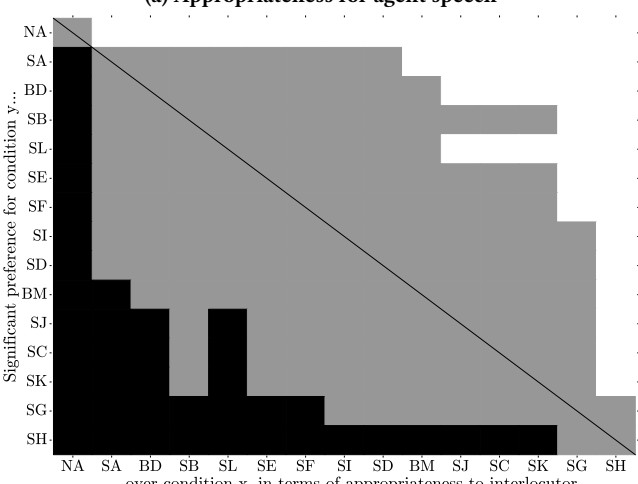

**(b) Appropriateness for interlocutor**

**Figure 3: Significant differences between conditions in the two appropriateness studies**

Figure 3 shows the pairwise significance in appropriateness study. White means the conditions listed on $y$-axis achieved an MAS significantly above the condition on the $x$-axis, black means the opposite ($y$ scored below $x$), and grey means no statistically significant difference at level $a = 0.05$ after correction for the false discovery rate. Our entry **SL** is rated significantly below or equal to other entries by appropriateness for speech. On the other hand, our solution's appropriateness to the interlocutor's speech is significantly below only natural motion **NA**.

## B RATING DISTRIBUTION AND PAIRWISE SIGNIFICANT DIFFERENCE FOR HUMAN-LIKENESS STUDY

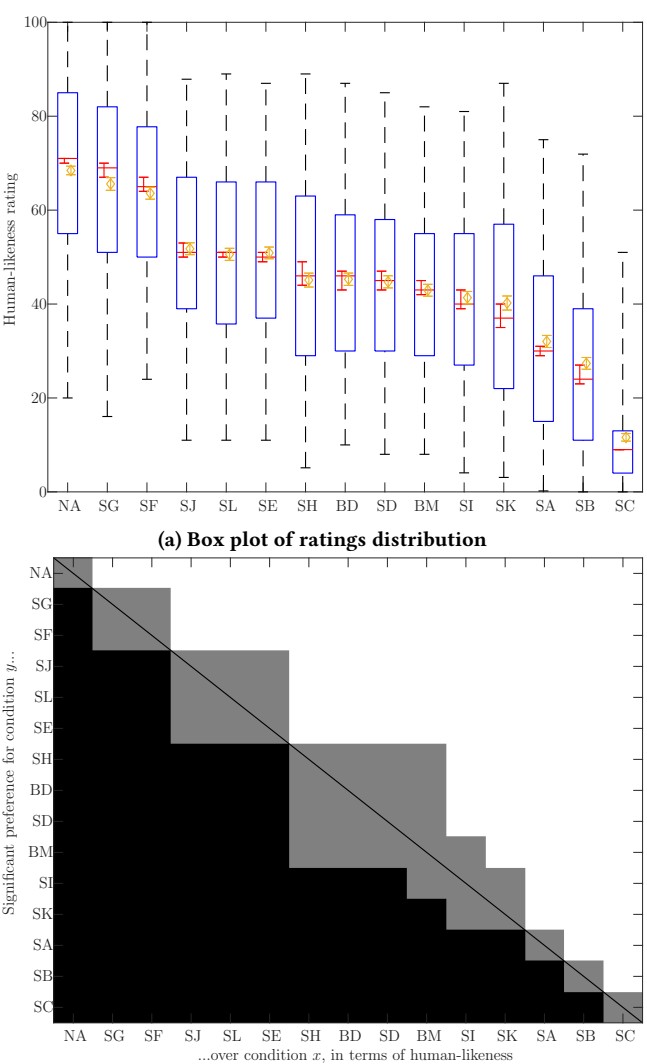

(a) Box plot of ratings distribution

(b) Significance of pairwise differences

**Figure 4: Visualisations of human-likeness study**

Figure 4 visualizes results of human-likeness study: 4a visualizing the rating distribution and 4b shows the pairwise significance.

In 4a Red bars are the median ratings (each with a 0.05 confidence interval); yellow diamonds are mean ratings (also with a 0.05 confidence interval). Box edges are at 25 and 75 percentiles, while whiskers cover 95% of all ratings for each condition. In 4b designation like in 3. Our entry **SL** is rated significantly below only natural motion **NA** and two participants' entries: **SG** and **SF**. It has also no significant difference from two other models: **SE** and **SJ**.