# OpenReview forum: "The FineMotion entry to the GENEA Challenge 2023: DeepPhase for conversational gestures generation"
_ACM.org/ICMI/2023/Workshop/GENEA_Challenge — GENEA Challenge 2023 Mainproceeding_

### Official Review · Reviewer_vtL1 · 2023-07-29
**The paper utilizes techniques such as VQ-VAE codebook encoding and phase features to improve the accuracy and human perceptibility of generated gestures, and shows innovation and potential by considering interlocutor information. Although lacking in more detailed experimental analysis, I recommend accepting the paper after revisions.**

**Rating:** 6
**Confidence:** 5

**Review:**

Abstract:

The authors present an innovative method that utilizes DeepPhase embeddings and VQ-VAE codebook encoding to generate stable and realistic gestures suitable for conversational agents. The authors employ the Periodic AutoEncoder from DeepPhase to generate gestures and leverage VQ-VAE codebook encoding to extract cyclic properties, facilitating interaction between the conversational agent and the speaker. Furthermore, the authors evaluate their method in the GENEA Challenge 2023 with promising results.

Review Feedback:

（1）Does the Interlocutor feature used in the article only exist in the codebook embedding? What if there is no Interlocutor information available?

（2）How is the gating network trained to obtain experts weights? How should we understand the mentioned experts weights and what is their significance? How does this branch affect the final results?

（3）How are the input experts weights, control vector, and Character state fused in the Motion Prediction Network?

（4）How do different Interlocutor control signals affect the generation of gestures for the speaker?

（5）In Table 1, the experimental results for Agent speech are similar to random results, suggesting that the model's learning ability is mediocre without the codebook feature. Have you considered using techniques like diffusion as the generation backbone network for better results?

I hope the above feedback can assist you in improving your work.

---

### Official Review · Reviewer_cEco · 2023-08-02
**Review of DeepPhase for conversational gestures generation**

**Rating:** 7
**Confidence:** 3

**Review:**

The proposed method adapts DeepPhase embedding and  VQ-VAE codebook encoding for conversational gestures generation.

It is intersting to represent motion's data with the DeepPhase. Unfortunately, with an average human-likeness of approximately 50%, and the agent's speech appropriateness performing worse than chance, the proposed approach lacked the capability to generate natural gesture motions effectively.

The absense of synthesized gesture videos makes it unclear to assess the performance accurately.

The results section lacks a comprehensive and detailed discussion of the findings. It remains unclear why the model yields poor appropriateness results. Adding an ablation study could really help.

Reproducibility: Lines 80-81
The authors release the code.


**Nominate For A Reproducibility Award:**

The authors release the code.

---

### Official Review · Reviewer_ztCh · 2023-08-03
**A valuable adaptation of an important motion synthesis method to gestures**

**Rating:** 7
**Confidence:** 5

**Review:**

This paper describes a gesture generation model that closely follows the DeepPhase approach, consisting of two parts:

1. Training a Periodic Autoencoder that transforms motion in joint velocity space to a low-dimensional phase manifold.
2. Training a motion generation model using the learned phase features and the task-specific conditioning - in this case, both agents' speech (GloVe embeddings and MFCCs) and the interlocutor's movements (encoded with a separate VQ-VAE). These control variables are further encoded using a bi-directional GRU.

The generator's input is a 2-second window containing: 1) character state (joint rotations and velocities), 2) the encoded control variables, and 3) motion phases for a random subset of frames. The network is trained to predict the motion features in the next frame based on the current frame only, while synthesis is done in an autoregressive fashion.

The idea of using DeepPhase for gesture synthesis is definitely interesting. Although much of the work is based on the original DeepPhase paper, there are some nontrivial modifications, such as the ``Control Variables Encoder'' and the phase blending during inference. The results in the human-likeness study indicate that it might be worthwile to further build on this line of work. I also think negative results, such as the failure to train the PAE on joint angles, or the low specificity to speech in the outputs, are particularly insightful. *Overall, I think this paper is a valuable contribution to the gesture synthesis community.* I would encourage the authors to further share their results online, e.g., by open-sourcing their codebase (if possible) and uploading rendered videos.

---
I have the following suggestions for the authors:
1) The abstract states that the motion is suitable for the interlocutor's behaviour. I personally find this statement too strong, since the observed effect sizes in that study seem miniscule compared to natural motion.
2) It would be valuable to describe the training run in terms of the number of optimisation steps instead of the number of epochs. The latter is not as useful without knowing the batch size.
3) It would be valuable to duplicate the "Condition" column of Table 1 (shortening the label to C.) so that the results of the two studies can be ordered separately. Alternatively, if space constraints allow, the two tables could be fully separated.
4) Below I am listing some of the typos I found during my review.
* line 42: missing full stop
* line 165: with [the] obtained latent representation
* line 131: the architecture utilizes [a] temporal
* line 252: >the more< the control variables vector would become ->   >the larger<
* lineTable 1 caption: Summary statistics of >appropriateness study< -> ... of >the appropriateness studies<

**Nominate For A Reproducibility Award:**

If the code is released, this paper might be a good candidate for the reproducibility award.

---

### Decision · Program_Chairs · 2023-08-04

**Decision:**

Accept (Main proceeding)

**Comment:**

Congratulations! All three reviewers recommended acceptance for this paper. The chairs have decided to accept the paper for publication in the main ICMI proceedings. For the camera-ready version, please consider including qualitative results, for example, by linking to a video.